# Dental follicle mesenchymal stem cells ameliorated glandular dysfunction in Sjögren's syndrome murine model

**Deniz Genç** [1,2]*, **Osman Bulut**[3], **Burcu Günaydin**[4], **Mizgin Göksu**[5], **Mert Düzgün**[5], **Yelda Dere**[6], **Serhat Sezgin**[7], **Akın Aladağ**[7], **Aziz Bülbül**[3]

**1** Faculty of Health Sciences, Muğla Sıtkı Koçman University, Muğla, Turkey, **2** Research Laboratories Center, Muğla Sıtkı Koçman University, Muğla, Turkey, **3** Milas Veterinary Medicine Faculty, Muğla Sıtkı Koçman University, Muğla, Turkey, **4** Department of Histology and Embryology, Institute of Health Sciences, Muğla Sıtkı Koçman University, Muğla, Turkey, **5** Faculty of Science, Department of Molecular Biology and Genetics, Muğla Sıtkı Koçman University, Muğla, Turkey, **6** Faculty of Medicine, Department of Pathology, Muğla Sıtkı Koçman University, Muğla, Turkey, **7** Faculty of Dentistry, Muğla Sıtkı Koçman Üniversity, Muğla, Turkey

* denizgenc@mu.edu.tr

**Data Availability Statement:** The anonymized data are freely accessible via DataverseNL using the following DOI: https://doi.org/10.34894/LZKWBR.

# Abstract

## Objective

Dental mesenchymal stem cells (MSCs) are potential for use in tissue regeneration in inflammatory diseases due to their rapid proliferating, multilineage differentiation, and strong anti-inflammatory features. In the present study, immunoregulatory and glandular tissue regeneration effects of the dental follicle (DF)MSCs in Sjögren's Syndrome (SS) were investigated.

## Methods

Dental follicle (DF) tissues were obtained from healthy individuals during tooth extraction, tissues were digested enzymatically and DFMSCs were cultured until the third passage. DFMSCs were labeled with Quantum dot 655 for cell tracking analysis. The induction of the SS mouse model was performed by the injection of Ro60-273-289 peptide intraperitoneally. DFMSCs were injected intraperitoneally, or into submandibular, or lacrimal glands. Splenocytes were analyzed for intracellular cytokine (IFN-γ, IL-17, IL-10) secretion in T helper cells, lymphocyte proliferation, and B lymphocyte subsets. Histologic analysis was done for submandibular and lacrimal glands with hematoxylin-eosin staining for morphologic examination.

## Results

The systemic injection of DFMSCs significantly reduced intracellular IFN-γ and IL-17 secreting CD4+ T cells in splenocytes (p<0.05), and decreased inflammatory cell deposits and fibrosis in the glandular tissues. DFMSCs differentiated to glandular epithelial cells in submandibular and lacrimal injections with a significant reduction in lymphocytic foci. The results showed that few amounts of DFMSCs were deposited in glandular tissues when

**Funding:** This study is funded by THE SCIENTIFIC AND TECHNOLOGICAL RESEARCH COUNCIL OF TURKEY (TUBITAK) with the Project number of 120S178. The funders had no role in study design, data collection, and analysis, decision to publish, or preparation of the manuscript.

**Competing interests:** The authors have no conflict of interest.

applied intraperitoneally, while high amounts of DFMSCs were located in glandular tissues and differentiated to glandular epithelial cells when applied locally in SS murine model.

## Conclusion

DFMSCs have the potential for the regulation of Th1, Th17, and Treg balance in SS, and ameliorate glandular dysfunction. DFMSCs can be a beneficial therapeutic application for SS.

## Introduction

Sjögren's Syndrome (SS) is a systemic autoimmune disease in which lymphocytic infiltration of salivary and lachrymal glands generally leads to xerostomia and xerophthalmia, and systemic complications develop in one-third of patients [1]. T helper 1 (Th1), Th17, and B cells have been reported to be involved in the pathogenesis of SS [2]. SS can be classified as; i) primary Sjögren's syndrome (pSS) that occurs in the absence of another underlying rheumatic disorder, and ii) secondary Sjögren's syndrome (sSS) that is associated with another underlying rheumatic disease, such as systemic lupus erythematosus (SLE), rheumatoid arthritis (RA), or scleroderma [3]. Current treatment of SS is carried out with biological agents and traditional disease-modifying drugs, but it is still difficult due to the complex pathogenesis of the disease and side effects of conventional drugs [4]. Therefore, it is crucial to develop new treatment approaches that target inflammatory responses and contribute to tissue regeneration in the treatment of SS disease.

Mesenchymal stem cells (MSCs) are adult multipotent cells originating from mesodermal and ectodermal lineages, which have the capacity of self-renewal and can differentiate into osteoblasts, adipocytes, and chondrocytes [5]. MSCs can be isolated from many adult tissues such as bone marrow, adipose tissue, umbilical cord, and dental tissues. Among them, dental MSCs are potent candidates for the treatment of inflammatory diseases due to an easily accessible source, easy to isolate, rapidly proliferating in the culture, multipotent differentiation ability, and high immunomodulatory effects [6]. However, there are limited studies on the effects of MSCs in the treatment of SS, and the effects of dental MSCs on lymphocyte phenotype, immunoregulation, and tissue regeneration in SS disease remain to be elucidated.

In the present study, we investigated the immunomodulatory and regenerative effects of DFMSCs on the glandular tissues of the murine model of SS by injecting them intraperitoneally or into the submandibular or lacrimal glands.

## Materials and method

In this study, the ethical approval was obtained from The Muğla Sıtkı Koçman University Animals Ethics Committee (24/07/2020.01) for animal studies and Muğla Sıtkı Koçman University Clinical Ethical Committee (08/08/2019.06) for collection of human dental tissues.

### Animals and study groups

Fifty 6–8 weeks old male Balb-c mice were purchased from the Experimental Animal Center of the Muğla Sıtkı Koçman University. Mice were housed in animal cages at the same center, and all experiments were approved by the ethics committee for experimental animals. The total sample size for the study is determined with G-power analysis, and calculated as n = 50

for the comparison of 5 groups including SS, Control, and dental-MSCs treatment groups when the effect size is 0.55, α-error is 0.05, power is 0.85 calculated by F tests [7]. Groups were determined as following; Group 1: Control group (n = 10), Group 2: SS Model (n = 10), Group 3a: intraperitoneal DFMSCs treatment group (n = 10), Group 3b: submandibular DFMSCs treatment group (n = 10), Group 3c: lacrimal DFMSCs treatment group (n = 10). Animals were sacrificed 28 days after treatment protocols. Experimental design is given in S1 Fig.

## Sjögren's syndrome experimental model

The induction of the SS murine model was performed as previously described [8]. In brief, Ro60-273-289 peptide with sequence LQEMPLTALLRNLGKMT was purchased from Biosynthesis, USA. The intraperitoneal injection was done with 50 μg of Ro60 peptide emulsified in 100 μL of Freuds' complete adjuvant (FCA) on day 1. The following immunizations were carried out on days 14, 36, 63, and 119 of the protocol with 50 μg Ro60 peptide in 100 μL of incomplete Freuds' adjuvant (iFA) per mouse. Control animals were immunized with FCA or iFA.

## Isolation and characterization of dental MSCs

Dental follicle tissues were obtained from 6 healthy subjects aged between 19–25 years, who applied to the Muğla Sıtkı Koçman University Dental Hospital for wisdom tooth extraction. Written consent was obtained from all participants according to the ethical guidelines. Dental tissues were mechanically and enzymatically digested as described previously [9]. In brief, dental tissues were cut into 0.5–1 mm pieces with a scalpel. The collagenase type I solution was used (3mg/mL in PBS) for the enzymatic digestion of tissues, at 37°C for 45 minutes. After the incubation period, cells were washed with 5 mL of DMEM supplemented with 10% FBS and centrifuged at 1500 rpm for 5 minutes. The remaining pellet was suspended in 5 mL DMEM supplemented with 10% FBS and 1% Penicillin/Streptomycin (100 U/mL) and cultured in a T25 flask until reaching 80–90% confluence. Dental MSCs in the third passage were analyzed for the positive cell surface markers of CD29, CD73, CD90, and CD105, and negative cell surface markers of CD14, CD34, and HLA-DR by staining the cells with anti-CD73 (APC), anti-CD90 (PerCp), anti-CD105 (FITC), anti-CD14 (PE), anti-CD34 (APC), anti-CD45 (PerCp), anti-HLA-DR (FITC) antibodies. All antibodies were purchased from BD Biosciences, US. Analysis was performed via flow cytometry (BD Accuri C6 Plus, BD Biosciences, US). The data was recorded as mean fluorescence intensity (MFI).

## Fluorescence labeling of dental MSCs

Dental MSCs were labeled with Quantum Dot 655 kit (Thermofisher, US) for homing analysis of MSCs in the murine model. Qdot labeling was performed as described in the manufacturer's protocol. In brief, component A and component B in the kit were mixed and incubated in the dark at room temperature for 5 minutes. $1 \times 10^6$ cells suspended in 100 μL of DMEM transferred into component A and B vial, vortexed for 30 seconds, and incubated for 60 minutes at 37°C in the dark chamber. Labeled cells were used within 2 hours for the treatment groups.

## Treatment of SS model mice with DFMSCs

The injections of DFMSCs were done with three injection ways as described before [10–12] with some modifications as follows; I) İntraperitoneal injection: $1 \times 10^6$ cells suspended in 0.5 mL of PBS was injected intraperitoneally with a 28-gauge needle [10]. II) Submandibular gland injection: $1 \times 10^6$ cells suspended in 0.1 mL PBS, and injected in the submandibular gland

below the throat line of the mouse with a 29-gauge needle [11]. III) Lacrimal gland injection: $1x10^6$ cells suspended in 20 μL PBS was injected in the extra orbital lacrimal gland through the front of the ear line with a 27-gauge needle [12]. All injections were done on day 126. Animals were sacrificed on the 28th day after DFMSCs treatment, on day 154th of the experimental protocol.

## Saliva and tears flow rate

Mouse saliva was collected according to a previous study [13]. Saliva and tears secretion was stimulated with an intraperitoneal injection of 0.05 mg of pilocarpine/100 g body weight. Total saliva was obtained from the oral cavity over 10 minutes using capillary tubes. Tears were collected for 10 seconds with a cotton thread containing phenol red at 5–10 minutes following the injection of pilocarpine hydrochloride [14]. The saliva and tears secretion rate measurement was done on days 126 and 154. The data for saliva flow rate was represented as microliters (μL)/10 minutes, and the data for tears rate was recorded as mm/10 seconds.

## Lymphocyte proliferation analysis

To evaluate the immunosuppressive effect of DFMSCs we carried out a proliferation analysis for lymphocytes, by staining splenocytes with Carboxyfluorescein succinimidyl ester (CFSE), as described before [15]. For the analysis of lymphocyte proliferation, 5 μM of CFSE (Thermofisher, US) was used to stain every $1x10^6$ cells in 1 mL PBS. The cell suspension was incubated in the dark at room temperature for 5 minutes, then mixed with 5 mL of cRPMI (RPMI 1640 + 10% FBS + 1% penicillin/streptomycin) and centrifuged at 2000 rpm for 5 minutes. After centrifugation, the supernatant was discarded, the remaining cells were suspended with cRPMI ($5x10^5$ cells/500 μL), and stimulated with anti-CD3 and anti-CD28 for specific T lymphocyte responses for 72 hours. Separately, B lymphocytes were stimulated with lipopolysaccharide (LPS) (1 μg/mL) for 6 days for the analysis of B cell proliferation rate. At the end of the culture period, the cells were stained either with anti-CD3 (APC), anti-CD4 (PE) to gate T lymphocytes or with anti-CD3 (APC) and anti-CD19 (PerCp) to gate B lymphocytes for 30 minutes at +4˚C, and CD3+CD4+ T cells or CD19+ B cells were gated in the flow cytometry analysis and analyzed for CFSE signaling in the FL-1 channel. All antibodies were purchased from BD Biosciences, US. The data was recorded as a percentage (%) in the selected lymphocyte population. The gating strategy is shown in S2 Fig.

## Immunophenotyping analysis

The spleens were removed through an incision from each of the mice after sacrification on day 154. As described before, the immunophenotyping analysis for Th1, Th2, Th17, and T regulatory (Treg) cells from splenocytes were done [16]. The spleens were cut into small pieces about 0.5 mm in diameter with a scalpel. Cells arising from the spleen were washed with 10 mL of RPMI 1640 medium, filtered through a 70 μm nylon filter, and then centrifuged at 200xg for 10 minutes. The cell pellet was counted by a hemocytometer, and $1x10^6$ cells/mL were stimulated with anti-CD3 and anti-CD28 (1 μg/ml) at 37˚C in a humidified 5% CO2 atmosphere for 72 hours. After the culture period splenocytes were analyzed for intracellular cytokine secretions. In brief, cells were washed twice with phosphate buffer solution (PBS) and centrifuged at 200xg for 5 minutes. The remaining pellet was stained with anti-mouse-CD3 and anti-mouse-CD4 to gate Th cells. Then, the cells permeabilized with a permeabilization solution for 40 minutes. The permeabilized cells were then stained with anti-IFNγ, anti-IL4, anti-IL17, or anti-IL10 antibodies for intracellular cytokine detection via flow cytometry (Accuri C6 Plus, BD Biosciences, US). B lymphocyte subsets were analyzed for plasma B cells and naïve B cell

populations with the stimulation of lipopolysaccharide (LPS) (1 μg/mL) for 6 days. CD19+-IgD-CD27+CD38+ cells were analyzed for plasma B lymphocytes. CD19+CD27-IgD+ cells were analyzed for naïve B lymphocytes. Cells were stained with anti-mouse-CD19 (APC), anti-mouse-CD27 (PE), anti-mouse-CD38 (FITC), anti-IgD (PerCp). FoxP3 expressing CD4+ T cells were also analyzed for T regulatory cell frequency. The splenocytes ($1x10^6$ cells/mL) were stimulated with anti-CD3 and anti-CD28 (10 μg/ml) at 37˚C in a humidified 5% CO2 atmosphere for 72 hours. After the culture period, cells were stained with anti-mouse-CD4 (FITC), anti-mouse-CD25 (APC) antibodies for cell surface markers. Cells were permeabilized and stained with anti-mouse-FoxP3 (PE) antibodies for the intracellular expression of FoxP3. All antibodies were purchased from BD Biosciences, US. The data was represented as the percentage (%) of the cells. The gating strategy is shown in S2 Fig.

## Detection of secreted cytokine levels

The cytokine levels were measured in the culture supernatants of splenocytes, and in the collected saliva and tears as described before [9,17,18], with some modifications. In brief, the isolated splenocytes were suspended in RPMI 1640 medium supplemented with 10%FBS and 1% penicillin/streptomycin (100 U/ml,100μg/mL) and were stimulated with anti-CD3 and anti-CD28 at 37˚C in a humidified 5%CO2 chamber for 72 hours. After the culture period, the supernatant was collected and 50 μL of each of the samples was incubated with microbeads for 2 hours. The saliva samples (50 μL) were directly incubated with the microbeads for 2 hours. Tear cytokine concentrations were measured from the collected 4–5 μL of tear samples. The samples were diluted 1 in 8.333 in a final volume of 25 μL and a total of 1μL of each capture bead. Levels of IFN-γ, IL-17 and IL-10 were determined using a Cytometric Bead Array (CBA) Human Th1/Th2/Th17 kit (BD Biosciences, San Diego, CA, USA) according to the manufacturer's instructions. The data were given as pg/ml.

## Homing and differentiation analysis of DFMSCs

The glandular samples were divided into two parts. One of the parts was subjected to histochemical analysis and the second part of the tissues was analyzed for α-smooth muscle actin (α-SMA) within Qdot labeled cells to determine the differentiation of DFMSCs to glandular epithelial cells. We analyzed homing of DFMSCs applied with intraperitoneal, submandibular, and lacrimal injections on the 7th day and 28th day after injection in the submandibular and lacrimal gland tissues as described previously [19]. Fresh glandular tissues were sent immediately to the pathology laboratory for the frozen section. An adjacent tissue block was fixed in 10% neutral buffered formalin and paraffin-embedded. Frozen sections were freshly studied by Leica CM-1860 UV. Sections were stained with anti-smooth muscle actin (α-SMA) and DAPI antibodies to detect Qdot labeled cells in the glandular tissue. The images were scanned by a fluorescent microscope (Nikon TS-2 FL Trinocular Inverted Microscope 5.9 Megapixel Head DS-Fi3, Japan). Antibodies were purchased from Abcam, US.

The differentiation of DFMSCs was analyzed by staining the glandular epithelial cells with anti-α-SMA antibody within Qdot labeled cells via flow cytometry. The tissue digestion was performed as described before [20]. Fresh tissue samples were cut into 0.5–1 mm pieces and enzymatically digested with 2 mL of DMEM low glucose supplemented with 3mg/ml collagenase type I, 2.4 U/mL dispase solution, 8 U/mL DNAse type I solution, and 6 mM CaCl2 at 37˚C for 90 minutes. The enzymatic activation was blocked with DMEM supplemented with 15%FBS and washed twice with the same medium. The samples were centrifuged at 1,242 x g for 5 minutes. The cell pellet was stained with anti-α-SMA (FITC) antibody and analyzed via flow cytometry. The epithelial differentiation of dental MSCs was determined by analyzing the

double-positive cell population of α-SMA (FL-1 channel) and Qdot (FL-3 channel). The data was represented as the percentage (%) of the cells.

## Histopathologic analysis

Glandular tissues were fixed in 10% buffered formalin at room temperature for 24 hours. Then glandular tissues were embedded in paraffin and serially sectioned into 4–5 μm thick paraffin sections. The sections were stained with hematoxylin-eosin (Bio Optica, Italy) using standard protocols. Sections were observed under the light microscope (Olympus BX46, Japan). The focus score in the gland was determined by the number of lymphocytic foci [21].

## Statistical analysis

Differences between groups were analyzed using the GraphPad Prism 9.0 version (GraphPad Software, Inc., CA, USA). Data were given as mean (Mean) ± standard deviation (SD) (minimum-maximum) values in each group. In detail, the secreted cytokine levels were in pg/mL and given as mean±SD. Flow cytometry analysis for cytokine secreting cells, and B lymphocyte subsets were presented as percentage (%) of cell populations. Graphical presentation and statistical analysis were calculated using GraphPad Prism 8 software (Graphpad Software Inc., USA). Comparison of the data of more than two groups was done by one-way ANOVA test. Histopathologic data were analyzed by the Mann-Whitney U test. An unpaired Student's t-test was used to compare cytokine levels between groups. $P < 0.05$ values were considered statistically significant and $p < 0.01$-$p < 0.001$ values were considered as highly statistically significant.

# Results

## Characterization of DFMSCs

DF-MSCs showed MSCs characteristics in the third passage with the high expressions of positive markers of CD73 (DFMSCs: 96.3±0.9%), CD90 (DFMSCs: 96.7±1.0%) CD105 (DFMSCs: 95.6±0.4%). Dental MSCs lack the expression of CD14, CD34, CD45 and HLA-DR as negative markers (S3 Fig).

## The systemic application of DFMSCs suppressed peripheral IFN-γ and IL-17 secreting T lymphocytes and reduced symptoms in a murine model of SS

We compared the immunomodulatory effect of intraperitoneal, submandibular and lacrimal injections of DFMSCs in a mouse model of SS by analyzing lymphocyte proliferative responses, and T and B lymphocyte phenotypes within splenocytes. Total lymphocytes were gated from the SSC-FSC area of flow cytometry and analyzed for CD3+CD4+ T lymphocytes proliferation. The cytokine secreting cells were analyzed for CD4+IFNγ+ or CD4+IL17+ or CD4+IL10+ cells for Th1, Th17, or IL-10 secreting T cells, respectively. CD19+ B cells were analyzed for plasma and naïve B lymphocyte subsets to determine the modulatory effects of DFMSCs in the spleen mononuclear cells of the SS murine model. We also measured saliva and tears secretion rate in SS murine model before and after the DFMSCs treatment and compared the data within application days and groups.

CD3+CD4+ T lymphocyte proliferation rate was significantly high in SS mice (Group 2) (46.1±9.3%) compared to control group (Group 1) (9.6±2.2%) (p<0.001). The intraperitoneally injected DFMSCs (Group 3a) significantly decreased CD3+CD4+ T lymphocyte proliferation rate (19.7±7.6%) compared to Group 2 (p<0.01). Submandibular and lacrimal injections of DFMSCs tended to decrease the lymphoproliferative responses of CD3+CD4+ T lymphocytes (39.9±8.1% and 43.7±6.8%, respectively) compared to Group 2, but no significant

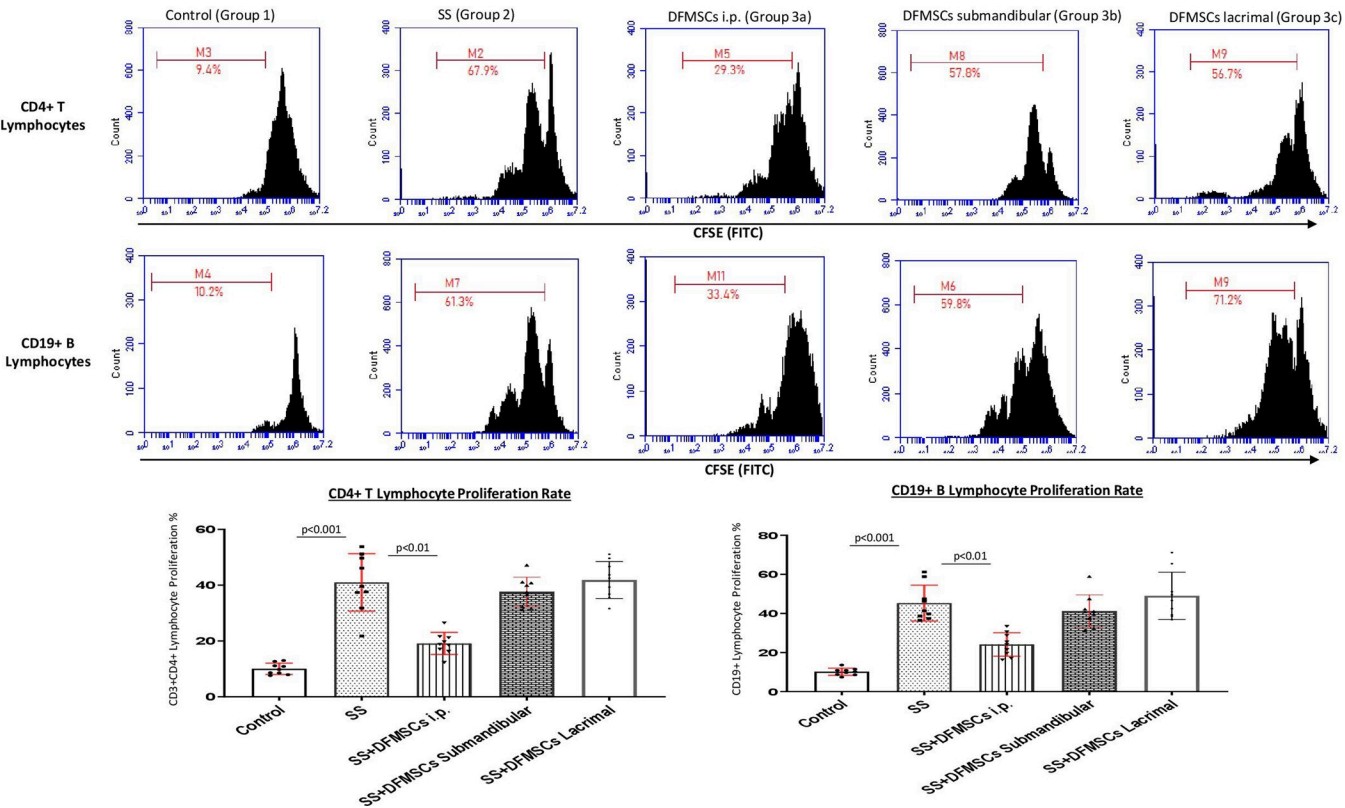

**Fig 1. Lymphocyte proliferation analysis in the splenocytes.** CD3+CD4+ T lymphocyte proliferation rate was significantly high in SS mice (Group 2) compared to the control group (Group 1) (p<0.001). The intraperitoneally injected DFMSCs (Group 3a) significantly decreased CD3+CD4+ T lymphocyte proliferation rate compared to Group 2 (p<0.01). Submandibular and lacrimal injections of DFMSCs tended to decrease the lymphoproliferative responses of CD3+CD4+ T lymphocytes compared to Group 2, but no significant difference was observed (p>0.05). CD19+ B lymphocyte proliferation rate was significantly high in SS mice (Group 2) compared to the control group (Group 1) (p<0.001). The intraperitoneal injection of DFMSCs (Group 3a) significantly decreased CD19+ B lymphocyte proliferation rate compared to Group 2 (p<0.01). Submandibular and lacrimal injections of DFMSCs did not significantly change the lymphoproliferative responses of CD19+ B lymphocytes compared to Group 2 (p>0.05).

difference was observed (p>0.05). CD19+ B lymphocyte proliferation rate was significantly high in SS mice (Group 2) (45.3±16.8%) compared to the control group (Group 1) (10.1 ±4.3%) (p<0.001). The intraperitoneally injected DFMSCs (Group 3a) significantly decreased CD19+ B lymphocyte proliferation rate (24.1±9.3%) compared to Group 2 (p<0.01). Submandibular and lacrimal injections of DFMSCs did not significantly change the lymphoproliferative responses of CD19+ B lymphocytes (46.9±12.9% and 53.3±17.9%, respectively) compared to Group 2 (p>0.05) **Fig 1**.

CD4+CD25+FoxP3+ T regulatory cell frequency was notably low in Group 2 (8.3±1.1%) compared to Group 1 (14.2±0.8%) (p<0.01). The intraperitoneal injection of DFMSCs significantly enhanced CD4+CD25+FoxP3+ T regulatory cell ratio (13.8±4.7%) when compared with Group 2 (p<0.01). The CD4+CD25+FoxP3+ T regulatory cell ratio in the intraperitoneal injected DFMSCs group (Group 3a) was close to Group 1, and no significant difference was observed between the two groups (p>0.05). The submandibular and lacrimal injections of DFMSCs did not significantly change CD4+CD25+FoxP3+ T regulatory cell frequency (9.1 ±0.6% and 8.7±0.9%) compared to Group 2 (p>0.05 and p>0.05, respectively) **Fig 2**.

CD4+IFN-γ+ T cell ratio in the splenocytes was significantly high in Group 2 (6.8±0.7%) compared to Group 1 (1.6±0.4%) (p<0.001). The intraperitoneal injection of DFMSCs significantly reduced CD4+IFN-γ+ T cell ratio (2.4±0.5%) compared to Group 2 (p<0.001). The

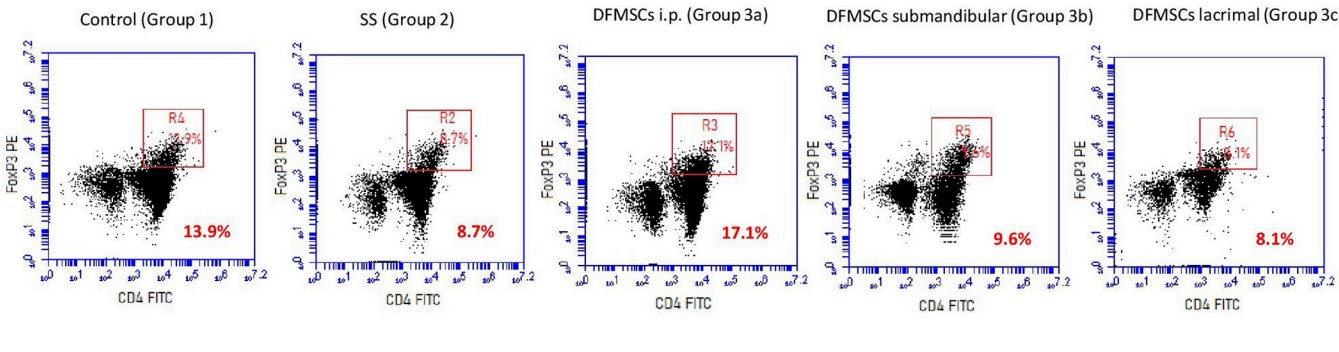

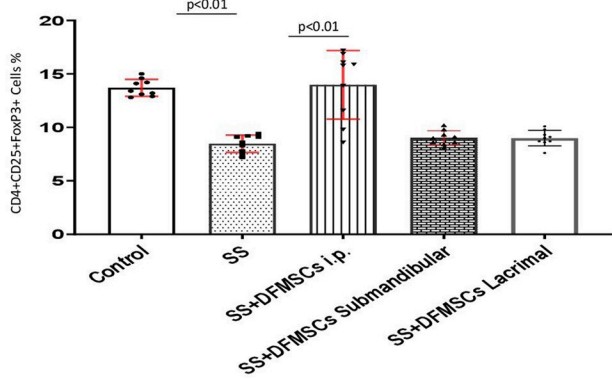

**Fig 2. FoxP3 expressing T regulatory cell frequency.** The intraperitoneal injection of DFMSCs significantly enhanced CD4+CD25+FoxP3+ T regulatory cell ratio compared to Group 2 (p<0.01). FoxP3 expressing CD4+CD25+ T regulatory cell ratio in intraperitoneal injected DFMSCs group (Group 3a) was close to Group 1, and no significant difference was observed between two groups (p>0.05). The submandibular and lacrimal injections of DFMSCs did not significantly change CD4+CD25+FoxP3+ T regulatory cell frequency compared to Group 2 (p>0.05 and p>0.05, respectively).

submandibular and lacrimal injected DFMSCs tended to decrease CD4+IFN-γ+ T cell ratio (6.2±0.4% and 6.7±0.6%) in the splenocytes, but no significant difference was observed when compared with Group 2 (p>0.05 and p>0.05). CD4+IL-17+ T cell ratio was significantly high in Group 2 (3.8±0.7%) compared to Group 1 (0.9±0.4%) (p<0.001). The intraperitoneal and submandibular injection of DFMSCs significantly reduced CD4+IL-17+ T cell ratio (1.4±0.6% and 2.0±0.3%, respectively) compared to Group 2 (p<0.001 and p<0.005, respectively). The lacrimal injected DFMSCs tended to decrease CD4+IL-17+ T cell ratio (3.5±0.3%) in the splenocytes, but no significant difference was observed when compared with Group 2 (p>0.05). CD4+IL-10+ T cell ratio was significantly low in Group 2 (2.2±0.3%) compared to Group 1 (3.4±0.8%) (p<0.05). The intraperitoneal injection of DFMSCs significantly increased CD4 +IL-10+ T cell ratio in Group 3a (5.2±0.6%) compared to Group 2 (p<0.01), but no significant difference was observed in submandibular and lacrimal injected DFMSCs (2.7±0.5% and 2.3 ±0.8%, respectively) when compared with Group 2 (p>0.05 and p>0.05, respectively) **Fig 3**.

The secreted cytokine levels in the culture supernatants of splenocytes were analyzed via flow cytometry. IFN-γ and IL-17 levels were significantly high in Group 2 (IFN-γ: 38.1±10.8 pg/mL, IL-17: 279.2±63.2 pg/mL) compared to Group 1 (IFN-γ: 21.3±7.1 pg/mL, IL-17: 199.8 ±23.4 pg/mL) (p<0.05 and p<0.01, respectively). IL-10 levels were significantly low in Group 2 (15.4±4.2 pg/mL) compared to Group 1 (21.8±9.7 pg/mL) (p<0.05). Intraperitoneal injection of DFMSCs significantly increased IL-10 levels in Group 3a (32.3±9.8 pg/mL) (p<0.01),

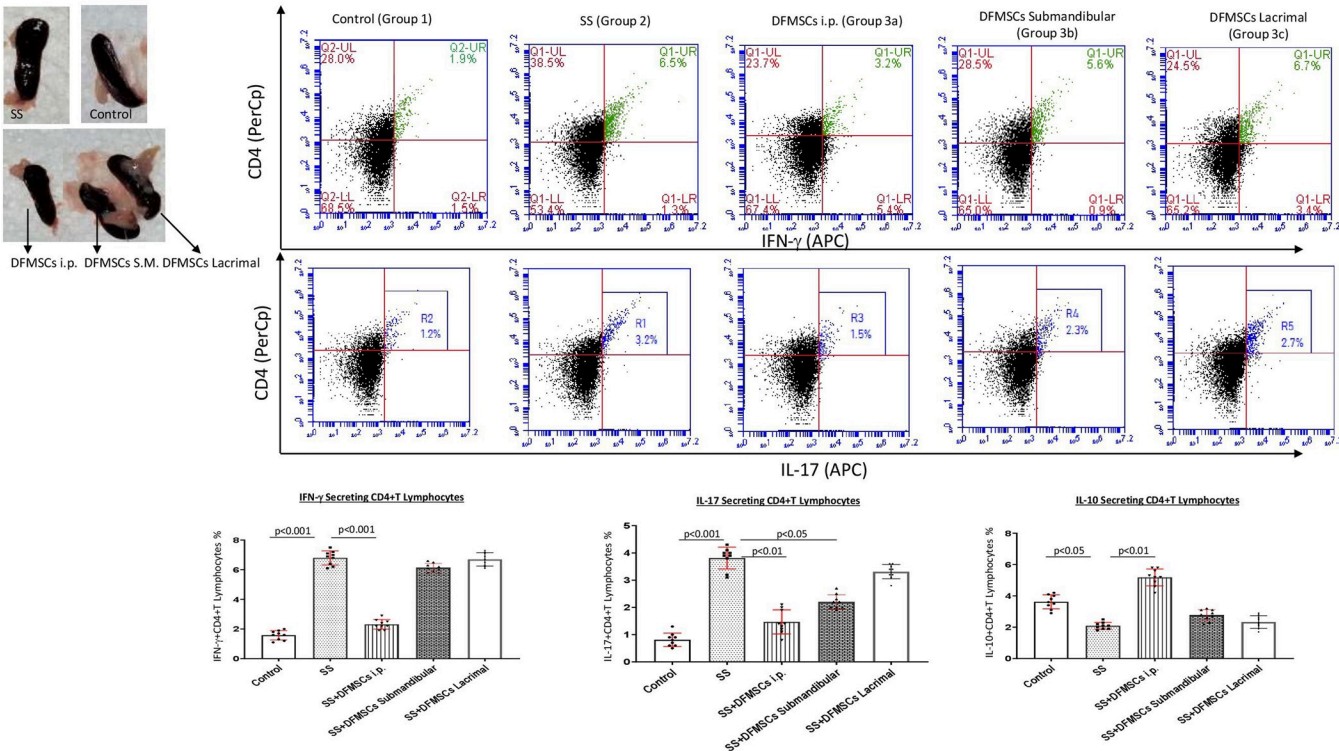

**Fig 3. Cytokine secreting CD4+T lymphocyte ratio.** The intraperitoneal injection of DFMSCs significantly reduced the CD4+IFN-γ+ T cell ratio compared to Group 2 (p<0.001). The submandibular and lacrimal injections of DFMSCs tended to decrease the CD4+IFN-γ+ T cell ratio (6.2±0.4% and 6.7±0.6%) in the splenocytes, but no significant difference was observed when compared with Group 2 (p>0.05 and p>0.05). The intraperitoneal and submandibular injections of DFMSCs significantly reduced the CD4+IL-17+ T cell ratio compared to Group 2 (p<0.001 and p<0.005, respectively). The lacrimal injected DFMSCs tended to decrease CD4+IL-17+ T cell ratio in the splenocytes, but no significant difference was observed when compared with Group 2 (p>0.05). The intraperitoneal injection of DFMSCs significantly increased CD4+IL-10+ T cell ratio in Group 3a compared to Group 2 (p<0.01), but no significant difference was observed in submandibular and lacrimal injected DFMSCs when compared with Group 2 (p>0.05 and p>0.05, respectively).

and significantly decreased the secreted IFN-γ and IL-17 levels (15.9±4.2 pg/mL and 203.7 ±19.2 pg/mL, respectively) compared to Group 2 (p<0.01 and p<0.01, respectively). Submandibular and lacrimal injections of DFMSCs did not significantly change the secreted IFN-γ (36.9±11.6 pg/mL and 34.7±7.8 pg/mL) (p>0.05 and p>0.05, respectively), IL-17 (265.2±57.4 pg/mL and 271.9±41.5 pg/mL) (p>0.05 and p>0.05, respectively), and IL-10 levels (12.8±3.5 pg/mL and 10.1±3.2 pg/mL) when compared with Group 2 (p>0.05 and p>0.05, respectively) **Fig 4A**.

The secreted IFN-γ and IL-17 levels in the saliva were significantly high in Group 2 (IFN-γ: 16.2±6.3 pg/mL, IL-17: 19.8±5.7 pg/mL) compared to Group 1 (IFN-γ: 8.2±2.1 pg/mL, IL-17: 2.9±1.2 pg/mL) (p<0.01 and p<0.001, respectively). There was no significant difference between Group1 (11.7±3.9 pg/mL) and Group 2 (10.8±1.7 pg/mL) in IL-10 levels in the saliva (p>0.05). Intraperitoneal injection of DFMSCs significantly increased IL-10 levels in Group 3a (16.5±3.8 pg/mL) (p<0.05), and significantly decreased the secreted IFN-γ and IL-17 levels (11.3±2.1 pg/mL and 12.1±2.4 pg/mL, respectively) compared to Group 2 (p<0.05 and p<0.05, respectively). Submandibular injection of DFMSCs significantly decreased the secreted IFN-γ and IL-17 levels in the saliva (8.3±1.9 pg/mL and 9.6±0.5 pg/mL, respectively) compared to Group 2 (p<0.01 and p<0.05, respectively). Submandibular injection of DFMSCs significantly increased IL-10 levels in the saliva (14.2±3.6 pg/mL) compared to Group 2 (p<0.05) **Fig 4B**.

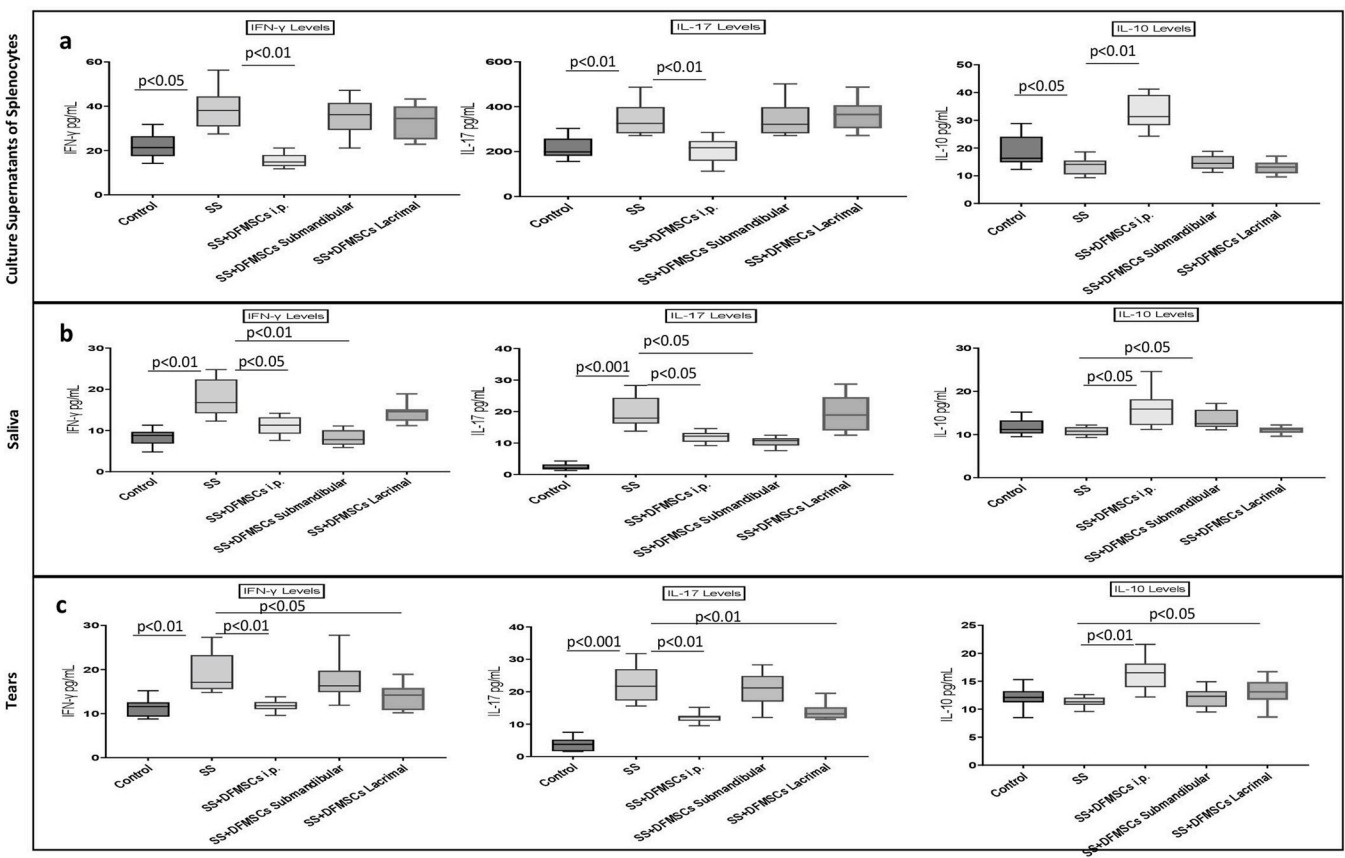

**Fig 4. The secreted cytokine levels in the culture supernatant of splenocytes, saliva, and tears.** (a) The cytokine levels in the culture supernatants of splenocytes. Intraperitoneal injection of DFMSCs significantly increased IL-10 levels in Group 3a (p<0.01), and significantly decreased the secreted IFN-γ and IL-17 levels compared to Group 2 (p<0.01 and p<0.01, respectively). Submandibular and lacrimal injections of DFMSCs did not significantly change the secreted IFN-γ (p>0.05 and p>0.05, respectively), IL-17 (p>0.05 and p>0.05, respectively), and IL-10 levels when compared with Group 2 (p>0.05 and p>0.05, respectively). (b) The cytokine levels in saliva. Intraperitoneal injection of DFMSCs significantly increased IL-10 levels in Group 3a (p<0.05), and significantly decreased the secreted IFN-γ and IL-17 levels compared to Group 2 (p<0.05 and p<0.05, respectively). Submandibular injection of DFMSCs significantly decreased the secreted IFN-γ and IL-17 levels in the saliva compared to Group 2 (p<0.01 and p<0.05, respectively). Submandibular injection of DFMSCs significantly increased IL-10 levels in the saliva compared to Group 2 (p<0.05). (c) The cytokine levels in tears. Intraperitoneal injection of DFMSCs significantly increased IL-10 levels in Group 3a (p<0.05), and significantly decreased the secreted IFN-γ and IL-17 levels compared to Group 2 (p<0.01 and p<0.01, respectively). Lacrimal injection of DFMSCs significantly decreased the secreted IFN-γ and IL-17 levels in the tears compared to Group 2 (p<0.05 and p<0.01, respectively). Lacrimal injection of DFMSCs significantly increased IL-10 levels in the saliva compared to Group 2 (p<0.05).

The secreted IFN-γ and IL-17 levels in the tears were significantly high in Group 2 (IFN-γ: 19.1±8.2 pg/mL, IL-17: 21.8±6.7 pg/mL) compared to Group 1 (IFN-γ: 11.9±3.1 pg/mL, IL-17: 4.8±1.1 pg/mL) (p<0.01 and p<0.001, respectively). There was no significant difference between Group1 (12.3±3.6 pg/mL) and Group 2 (11.6±2.1 pg/mL) in IL-10 levels in the tears (p>0.05). Intraperitoneal injection of DFMSCs significantly increased IL-10 levels in Group 3a (17.1±4.3 pg/mL) (p<0.05), and significantly decreased the secreted IFN-γ and IL-17 levels (12.1±2.6 pg/mL and 11.9±2.1 pg/mL, respectively) compared to Group 2 (p<0.01 and p<0.01, respectively). Lacrimal injection of DFMSCs significantly decreased the secreted IFN-γ and IL-17 levels in the tears (14.6±3.9 pg/mL and 12.3±4.2 pg/mL, respectively) compared to Group 2 (p<0.05 and p<0.01, respectively). Lacrimal injection of DFMSCs significantly increased IL-10 levels in the saliva (13.4±4.6 pg/mL) compared to Group 2 (p<0.05) **Fig 4C**.

The saliva flow rate was significantly low in Group 2 compared to Group 1 on day 126 (Group 1: 307±39 μL, Group 2: 143±47 μL, p<0.01) and on day 154 (Group 1: 301±54 μL,

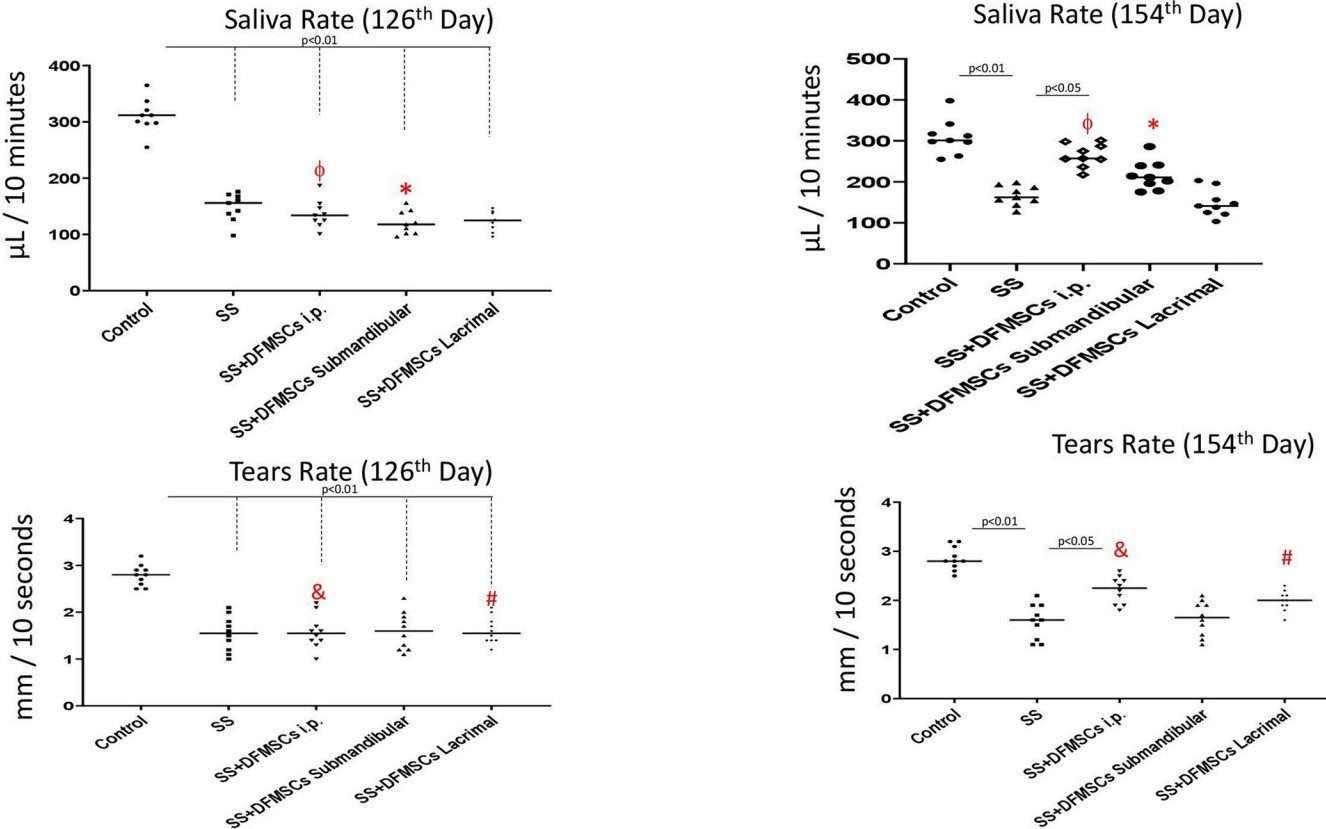

**Fig 5. Saliva and tears flow rate.** The intraperitoneal injection of DFMSCs significantly increased the saliva flow rate in Group 3a compared to Group 2 on day 154 (p<0.05) and increased the saliva flow rate in Group 3a on day 154 compared to day 126 (p<0.05). The lacrimal injection of DFMSCs in Group 3c did not significantly change the saliva flow rate in SS mice when compared with Group 2 on day 154 (p>0.05). The intraperitoneal injection of DFMSCs significantly increased tears rate in Group 3a compared to Group 2 on day 154 (p<0.05), and increased the tears rate in Group 3b on day 154 compared to day 126 (p<0.05).

Group 2: 158±32 μL, p<0.01). The intraperitoneal injection of DFMSCs significantly increased the saliva flow rate in Group 3a (247±35 μL) compared to Group 2 on day 154 (p<0.05) and increased the saliva flow rate in Group 3a on day 154 compared to day 126 (141±28 μL) (p<0.05). The lacrimal injection of DFMSCs in Group 3c did not significantly change the saliva flow rate in SS mice when compared with Group 2 on day 154 (137±23 μL) (p>0.05). The tears secretion rate was significantly low in Group 2 compared to Group 1 on day 126 (Group 1: 2.8±0.5 mm/10 seconds, Group 2: 1.4±0.6 mm/10 seconds), and on day 154 (Group 1: 2.8±0.4 mm/10 seconds, Group 2: 1.5±0.6 mm/10 seconds) (p<0.01). The intraperitoneal injection of DFMSCs significantly increased tears rate in Group 3a (2.2±0.5 mm/10 seconds) compared to Group 2 on day 154 (p<0.05), and increased the tears rate in Group 3b on day 154 compared to day 126 (1.9±0.3 mm/10 seconds) (p<0.05) **Fig 5**.

The ratio of CD19+IgD+CD27+ naïve B lymphocytes in the splenocytes were significantly low in Group 2 (19.9±5.8%) compared to Group 1 (27.2±4.9%) (p<0.05). Intraperitoneal injection of DFMSCs significantly increased the ratio of CD19+IgD+CD27+ naïve B lymphocytes in Group 3a (24.1±3.9%) compared to Group 2 (p<0.05). CD19+IgD-CD38+CD27+ plasma cell ratio was significantly high in Group 2 (3.6±1.2%) compared to Group 1 (0.6±0.4%) (p<0.01). Intraperitoneal injection of DFMSCs significantly decreased the ratio of CD19+IgD-CD38+CD27+ plasma cells in Group 3a (1.9±0.7%) compared to Group 2 (p<0.05). Submandibular and lacrimal injections of DFMSCs did not significantly changed

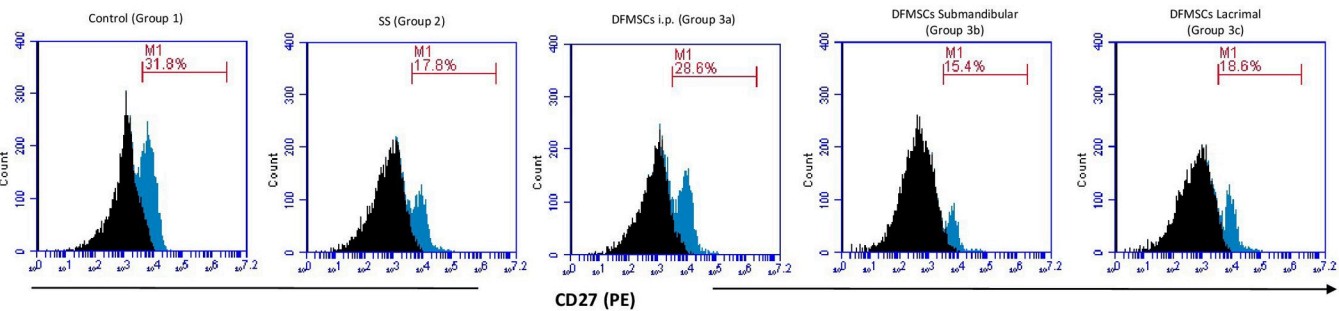

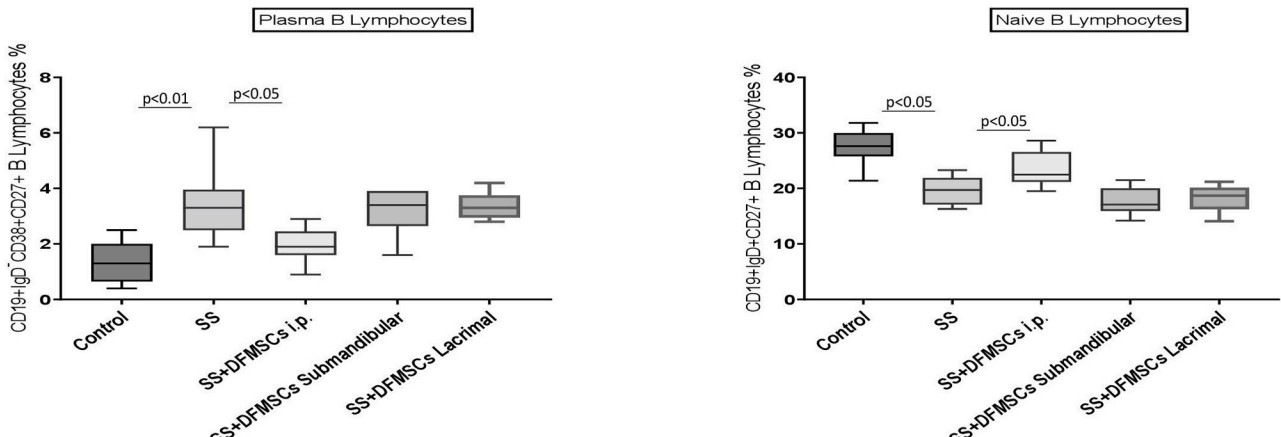

**Fig 6. Flow cytometry analysis of plasma and naïve B cell subsets in the splenocytes.** Intraperitoneal injection of DFMSCs significantly increased the ratio of CD19+IgD+CD27+ naïve B lymphocytes in Group 3a compared to Group 2 (p<0.05). CD19+IgD-CD38+CD27+ plasma cell ratio was significantly high in Group 2 compared to Group 1 (p<0.01). Intraperitoneal injection of DFMSCs significantly decreased the ratio of CD19+IgD-CD38+CD27+ plasma cells in Group 3a compared to Group 2 (p<0.05). Submandibular and lacrimal injections of DFMSCs did not significantly change CD19+IgD+CD27+ and CD19 +IgD-CD38+CD27+ cell ratio in Group 3b and Group 3c when compared with Group 2 (p>0.05).

CD19+IgD+CD27+ (Submandibular: 17.5±4.1%, Lacrimal: 18.2±2.9%) and CD19+IgD-CD38 +CD27+ cell ratio (Submandibular: 3.5±1.0%, Lacrimal: 3.4±0.8%) in Group 3b and Group 3c when compared with Group 2 (p>0.05) **Fig 6**.

## DFMSCs administered by intraperitoneal injection in SS mice migrated to submandibular and lacrimal glands in low amounts

The glandular tissues were analyzed for the migration and homing of DFMSCs after intraperitoneal, submandibular, and lacrimal injections at the 7[th] day and 28[th] days. The Qdot labeled cells in the tissue samples were analyzed by a fluorescent microscope. Results showed that Qdot labeled DFMSCs were located in submandibular and lacrimal glands within 7 days when injected locally. Intraperitoneally injected DFMSCs migrated preferentially to submandibular glands in the first 7 days and permanently localized as foci in the exocrine gland for up to 28 days **Fig 7**.

## Locally applied DFMSCs highly differentiated into glandular epithelial cells

The differentiation of Qdot labeled DFMSCs was analyzed by staining the glandular epithelial cells with α-SMA antibody. Qdot labeled cells were analyzed within glandular epithelial cells

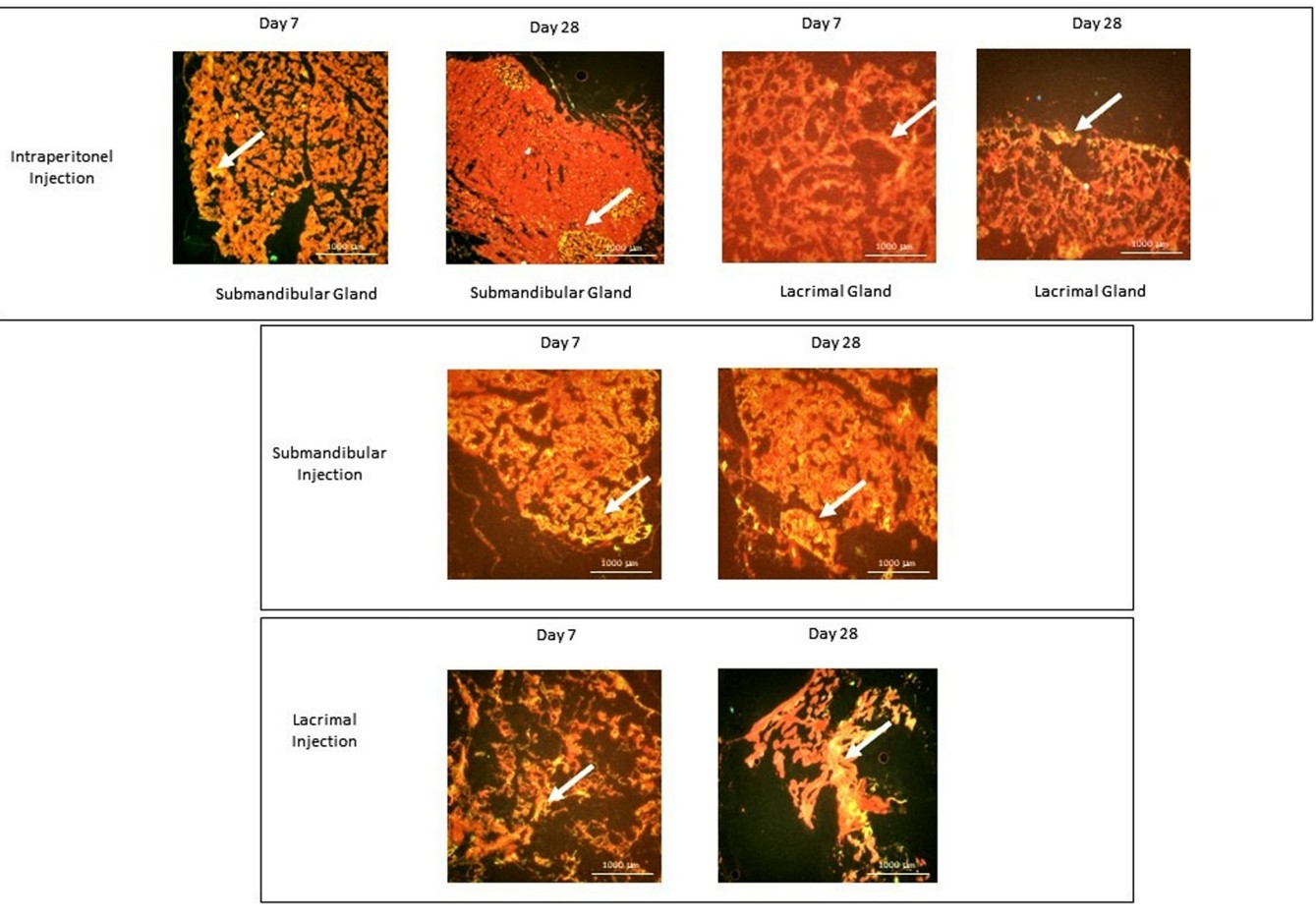

**Fig 7. Homing of DFMSCs in submandibular and lacrimal glands in SS mouse model.** Quantum dot 655 (Qdot-655) labeled cells were analyzed with a fluorescence microscope in the glandular tissues of the SS murine model on the 7th and 28th days after intraperitoneal, and local injections of DFMSCs. Qdot labeled DFMSCs were located in submandibular and lacrimal glands within 7 days when injected locally. Intraperitoneally injected DFMSCs migrated preferentially to submandibular glands in the first 7 days and localized as foci in the exocrine gland for up to 28 days.

by a fluorescent microscope. Submandibular and lacrimal injections of DFMSCs showed high differentiation into glandular epithelial cells on day 28 compared to the intraperitoneally injected DFMSCs. The data demonstrated that intraperitoneally injected DFMSCs act by regulating lymphocyte responses, not by differentiation into glandular epithelial cells in the SS mouse model **Fig 8A**.

Intraperitoneal, submandibular and lacrimal injected Qdot labeled DFMSCs were analyzed for α-SMA signaling via flow cytometry. The cells were analyzed in a double positive area for Qdot labeled cells (DFMSCs) in the FL-3 channel and α-SMA (FITC) in the FL-1 channel to determine DFMSCs expressing the epithelial marker. Results showed that submandibular and lacrimal injections of DFMSCs highly express α-SMA in submandibular (12.5±2.1%) and lacrimal (10.1±1.8%) glands of SS mouse model compared to intraperitoneal injections (Submandibular: 1.2±0.7%, Lacrimal: 0.9±0.4%) ($p<0.001$ and $p<0.001$, respectively) **Fig 8B**.

### DFMSCs reduced lymphocytic infiltration and fibrosis in glandular tissues

In the present study, we analyzed lymphocytic infiltration and tissue integrity by staining the sections with a hematoxylin-eosin stain. Glandular tissues of SS mice showed high amounts of

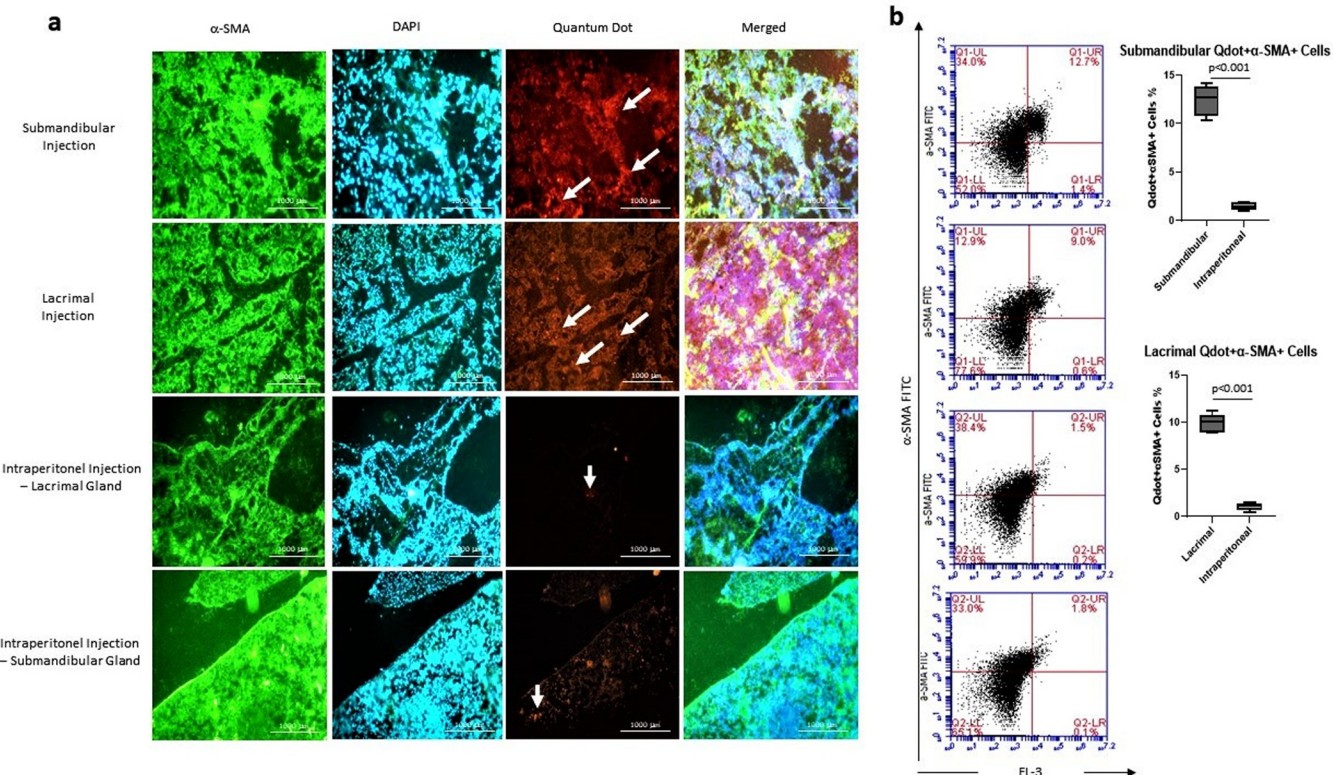

**Fig 8. The differentiation analysis of DFMSCs into epithelial lineage in glandular tissues.** (a) The fluorescent microscope images of glandular tissues. Glandular tissues were stained with α-SMA (FITC) and DAPI, and analyzed for Quantum dot 655 (Qdot-655) labeled cells. Submandibular and lacrimal injections of DFMSCs showed high differentiation into glandular epithelial cells on day 28 compared to the intraperitoneally injected DFMSCs. (b) Flow cytometry analysis for α-SMA stained cells within Qdot-655 labeled cells. The submandibular and lacrimal injections of DFMSCs highly express α-SMA in submandibular and lacrimal glands of SS mouse model compared to intraperitoneal injections (p<0.001 and p<0.001, respectively).

lymphocytic foci (Focus score >1) (number of lymphocytic foci in submandibular glands (mean±SEM): 4.6±0.5) (number of lymphocytic foci in lacrimal glands (mean±SEM): 3.4±0.6) compared to control subjects (Focus score <1) (number of lymphocytic foci in submandibular glands (mean±SEM): 0.2±0.8) (number of lymphocytic foci in lacrimal glands (mean±SEM): 0.4±0.6) (p<0.001). Also, tissue destruction and fibrosis in submandibular and lacrimal glands in Group 2 were observed in the glandular sections. DFMSCs reduced lymphocytic infiltrates in submandibular glands (number of lymphocytic foci (mean±SEM): 1.2±0.6) (p<0.01) and lacrimal glands (number of lymphocytic foci (mean±SEM): 0.8±0.4) (p<0.05), and ameliorated tissue destruction when applied intraperitoneally. Submandibular or lacrimal injection of DFMSCs significantly reduced lymphocytic infiltration and fibrosis in submandibular (number of lymphocytic foci (mean±SEM): 1.6±0.7) (p<0.05) or lacrimal glands (number of lymphocytic foci (mean±SEM): 1.4±0.5) (p<0.05) **Fig 9**.

## Discussion

Sjögren's syndrome (SS) is an autoimmune disease characterized by the destruction of the exocrine glands and lymphocyte infiltrates in the salivary and lacrimal glands [22]. Due to the potent immunomodulatory and regenerative effects, MSCs have been investigated in several autoimmune diseases such as rheumatoid arthritis, systemic lupus erythematosus, and Crohn's disease [23]. The studies demonstrated that MSCs exert modulatory effects on both innate and

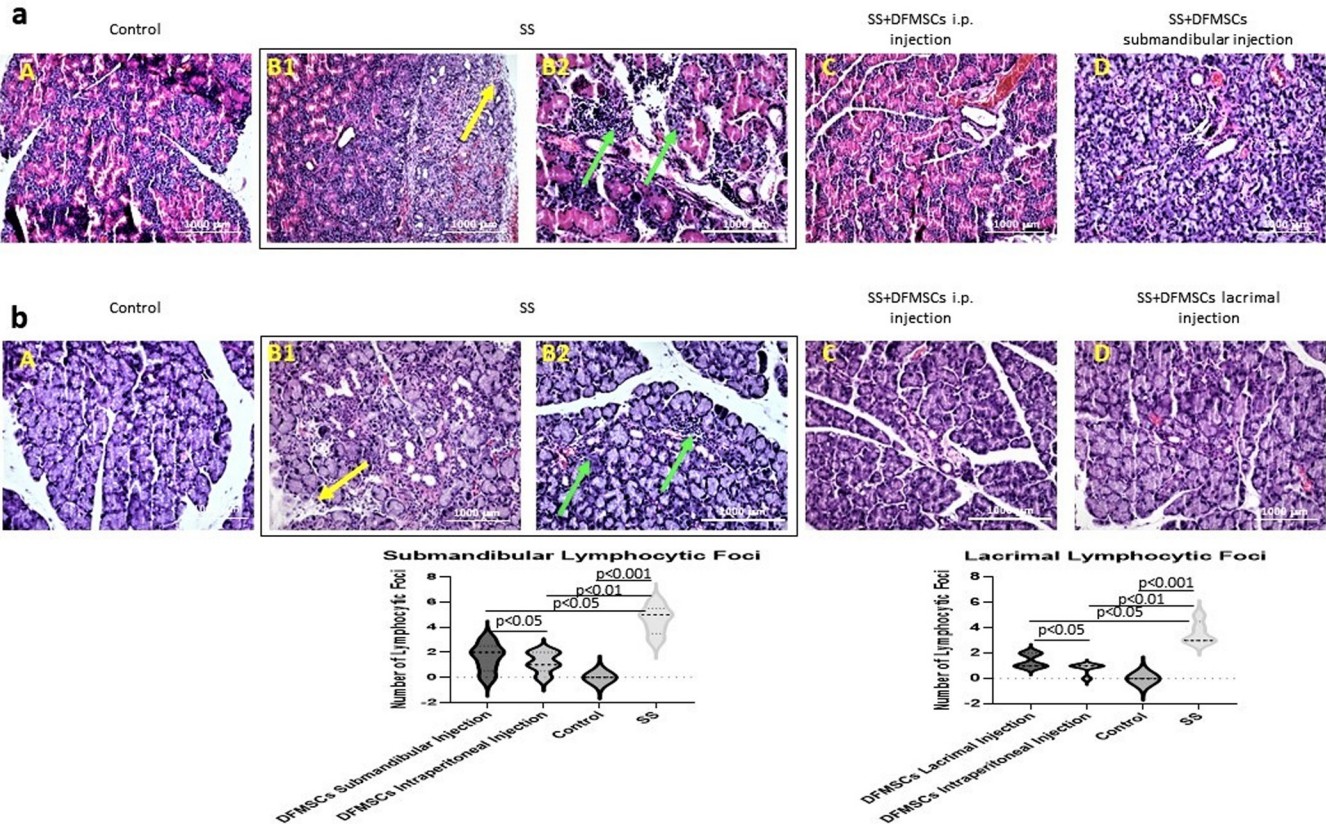

**Fig 9. Histopathological analysis of glandular tissues.** (a) Hematoxylin-eosin staining of submandibular and (b) lacrimal glands. The statistical analysis of lymphocytic foci of glandular tissues. Submandibular and lacrimal glands of SS mice showed high amounts of lymphocytic foci (Focus score >1) compared to control subjects (Focus score <1) (p<0.001). The lymphocytic foci were shown with green arrows. Tissue destruction and fibrosis in submandibular and lacrimal glands in Group 2 were observed in the glandular sections (yellow arrows). DFMSCs significantly reduced lymphocytic infiltrates in submandibular glands (p<0.01) and in lacrimal glands (p<0.05), and ameliorated tissue destruction when applied intraperitoneally. Submandibular or lacrimal injection of DFMSCs significantly reduced lymphocytic infiltration and fibrosis in submandibular (p<0.05) or lacrimal glands (p<0.05).

adaptive immune cells, by inhibiting the activation and plasticity of immune cells [24]. In the present study, we investigated the immunomodulatory and regenerative effects of DFMSCs, which can be isolated at a high amount in a culture condition and has strong anti-inflammatory and potent regenerative effects, on the SS murine model by systemic and local applications. We found that DFMSCs down-regulate immune responses reduce lymphocytic infiltration when administered by systemic injection, and can differentiate into glandular epithelial cells when administered into glandular tissue.

Previous studies have shown that bone marrow-derived MSCs upregulate the immunosuppressive effect of myeloid-derived suppressor cells by transforming the growth factor-b (TGF-β)/Smad pathway. In addition, PGE2 production by MSCs increases Foxp3+ T regulatory cell frequency [25–29]. However, given the complexity of pSS and the diverse immunosuppressive mechanisms of MSCs isolated from different tissues, uncovering new sources of MSCs that may have higher immunosuppressive capacity will contribute to the expansion of cellular therapy options [30]. In the present study, we demonstrated that systemic administration of DFMSCs strongly regulates Th1 and Th17 responses in splenocytes by enhancing the frequency of T regulatory cells and up-regulating IL-10 secreting CD4+ T cells, and by down-regulating plasma B cells in the favor of naïve B cells. The systemic injection also reduced lymphocytic infiltrate and tissue destruction in glandular tissues. After local injection,

DFMSCs differentiated into glandular epithelial cells in submandibular and lacrimal glands and reduced lymphocytic foci in glandular tissues. The improvement of SS manifestations was observed mainly in the systemic injection of DFMSCs, while a slight recovery was observed in local injections. Consistent with previous studies, the results of this study showed that local application of DFMSCs promotes regeneration in damaged glandular tissue and systemic administration of DFMSCs reduces immune responses in both exocrine glands and the periphery.

In this study, it was shown that intraperitoneal injection of DFMSCs down-regulated lymphoproliferative responses in splenic CD4+T and CD19+B cells, but failed to inhibit the proliferative responses of splenocytes at local injections. MSCs can suppress T-cell proliferation by secreting anti-inflammatory mediators such as transforming growth factor β1 (TGFβ1) and hepatocyte growth factor (HGF) [31,32]. To determine the regulatory effects of DFMSCs on the inflammatory mediators we further investigated the cytokine secretion in the splenocytes, saliva, tears, and splenic T and B lymphocyte subsets in the SS murine model.

Studies to date have demonstrated that IFN-γ and IL-17 are dominant cytokines found in the exocrine glands of SS patients and animal models of SS by enhancing T regulatory (Treg) and IL-10 secreting cells [33–41]. In this study, it was observed that systemic administration of DFMSCs suppressed the secretion of IFN and IL-17 cytokines in both splenocytes, saliva, and tears, by increasing the frequency of FoxP3-expressing T-regulatory cells (Treg) and the proportion of IL-10-producing lymphocytes. However, locally injected DFMSCs did not significantly change the amount of inflammatory cytokines. The data suggest that systemic administration of DFMSCs promotes peripheral tolerance in the favor of regulatory T lymphocyte phenotype.

Autoreactive antibody-producing plasma B cells are other cell types that are important in the pathogenesis of SS [42]. It is known that MSCs can suppress B cell proliferation and differentiation into plasma cells via IL-1Ra signaling, and can inhibit IgM production by B cells [43–45]. In the present study, the results showed that intraperitoneal injection of DFMSCs resulted in a significant reduction in the plasma B cell population in favor of naive B cells. However, no significant difference was observed in the ratio of naïve and plasma B cells in splenocytes of the SS mouse model with local injections of DFMSCs. These results suggest that locally administered DFMSCs may have a potent role in damaged tissue rather than peripheral immune responses.

In previous studies, it has been reported that high levels of IFN-γ, IL-17, IL-1RA, IL-4, and IL-2 were observed in tears of SS patients when compared with non-SS patients with sicca symptoms [35,46]. In the present study, we showed that local administration of DFMSCs into submandibular and lacrimal glands have also been effective in suppressing inflammatory cytokines in the saliva or tears that play a crucial role in the pathogenesis of SS, by enhancing IL-10 levels. Compatible with these results, the frequency of IFN-γ secreting Th1 cells and IL-17 secreting Th17 cells in the splenocytes strongly reduced with intraperitoneal administration of DFMSCs with the increased frequency of IL-10 secreting CD4+ T lymphocytes, while no significant difference was observed in the IFN-γ, IL-17 or IL-10 secreting cells in the splenocytes with the local administration of DFMSCs.

Histopathologic findings in SS include focal lymphocytic infiltrates in the exocrine glands. The infiltrate contains CD4+ Th cells, B cells, and plasma cells leading to glandular dysfunction that manifests as dry eyes and dry mouth [47–50]. We evaluated the symptomatic findings to determine the therapeutic efficacy of DFMSCs on SS. The flow rate of saliva and tears notably increased with systemic and local injections of DFMSCs. When evaluated together with the cytokine levels in saliva and tears, it is thought that DFMSCs may have a regenerative effect on

the damaged tissue due to the suppression of inflammatory responses in exocrine glands with locally applied DFMSCs.

The results of this study demonstrated that DFMSCs can migrate to exocrine glands when administered intraperitoneally. It has been observed that locally applied DFMSCs are permanently located in the exocrine glands. We, therefore, investigated the regenerative effects of DFMSCs with systemic and local administration by evaluating homing and differentiation ability of DFMSCs. It was observed that DFMSCs administered by intraperitoneal injection migrated to the exocrine tissue in the first 7 days and were localized as foci in the exocrine gland on the 28th day. It was demonstrated that locally injected DFMSCs were distributed in the exocrine glands in large numbers and permanently localized until the 28th day. After observing homing of DFMSCs, we performed histologically and flow cytometry analysis for the differentiation ability of Qdot labeled DFMSCs by staining tissue samples with α-SMA as an epithelial marker. The results indicated the differentiation ability of DFMSCs into glandular epithelial cells with local injections in the SS murine model. Previous studies demonstrated that human umbilical cord MSCs adapt to an epithelial phenotype when co-cultured with human salivary gland biopsies of SS patients, and bone marrow-derived MSCs showed epithelial differentiation potential when cultured in the conditioned media with keratinocyte growth factor, epidermal growth factor, hepatocyte growth factor, and insulin-like growth factor, which indicate that MSCs can differentiate into epithelial lineage under appropriate conditions [51,52]. In this study, we demonstrated the ability of DFMSCs to differentiate into an epithelial phenotype in a murine model of SS for the first time.

In this study, DFMSCs was shown to be an alternative to cell-based therapies by demonstrating their regulatory effects on T and B lymphocyte responses in SS. While DFMSCs substantially target peripheral immune responses when applied systemically, they prefer differentiating to epithelial cells when applied locally. The underlying mechanism of epithelial differentiation potential of DFMSCs in SS should be further investigated.

## Conclusion

In conclusion, we demonstrated the immunomodulatory effects of DFMSCs on T and B lymphocyte responses in a murine model of SS. In addition, DFMSCs have the potential to differentiate into glandular epithelial cells in the SS when applied locally. DFMSCs may be a potent candidate in cellular therapy options for SS.

## Supporting information

**S1 Fig. Study design.** The SS murine model was performed with the intraperitoneal injection of 50 μg of Ro60 peptide emulsified in 100 μL of Freuds' complete adjuvant (FCA) on day 1. The following immunizations were carried out on days 14, 36, 63, and 119 with 50 μg Ro60 peptide. DFMSCs are administered by intraperitoneal, submandibular, or lacrimal injections. (TIF)

**S2 Fig. Gating strategy for flow cytometry analysis.**
(TIF)

**S3 Fig. Characterization of DFMSCs.** DFMSCs expressed positive markers (CD73, CD90, and CD105) over 95%, and lack the expressions of negative markers (CD14, CD34, and HLA-DR). (TIF)

**S1 Raw data.**
(DOCX)

## Author Contributions

**Investigation:** Deniz Genç, Osman Bulut, Burcu Günaydin, Mizgin Göksu, Mert Düzgün, Yelda Dere, Serhat Sezgin, Aziz Bülbül.

**Methodology:** Deniz Genç, Yelda Dere, Akın Aladağ.

**Project administration:** Deniz Genç.

**Resources:** Deniz Genç, Serhat Sezgin.

**Supervision:** Akın Aladağ, Aziz Bülbül.

**Writing – original draft:** Deniz Genç.

**Writing – review & editing:** Aziz Bülbül.

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
