## [Decision Letter · Decision Letter 0]

27 Jan 2022

PONE-D-21-36398DENTAL FOLLICLE MESENCHYMAL STEM CELLS AMELIORATED GLANDULAR DYSFUNCTION IN SJÖGRENS’ SYNDROME MURINE MODELPLOS ONE

Dear Dr. Genç,

Thank you for submitting your manuscript to PLOS ONE. After careful consideration, we feel that it has merit but does not fully meet PLOS ONE’s publication criteria as it currently stands. Therefore, we invite you to submit a revised version of the manuscript that addresses the points raised during the review process.

This manuscript has been carefully evaluated by two experts in the field. It is of interest pending some alterations that are needed, as specified by each referee.The Authors must amen the manuscript following all the reviewers criticisms and highlight in the new text all the changes in red.A point to point rebuttal letter to all the points raised is also needed.

We look forward to receiving your revised manuscript.

Kind regards,

Gianpaolo Papaccio, M.D., Ph.D.

Academic Editor

PLOS ONE

Journal Requirements:

 “This study is funded by THE SCIENTIFIC AND TECHNOLOGICAL RESEARCH COUNCIL OF TURKEY (TUBITAK) with the Project number of 120S178.”

“This study is funded by THE SCIENTIFIC AND TECHNOLOGICAL RESEARCH COUNCIL OF TURKEY (TUBITAK) with the Project number of 120S178. We thank to Rüştü Kuşcu for his skillful assistance.“

“This study is funded by XXXXXXXX with the Project number of 120S178. We thank to XXXXXX for his skillful assistance.”

“This study is funded by THE SCIENTIFIC AND TECHNOLOGICAL RESEARCH COUNCIL OF TURKEY (TUBITAK) with the Project number of 120S178.”

Reviewers' comments:

Reviewer's Responses to Questions

**Comments to the Author**

1. Is the manuscript technically sound, and do the data support the conclusions?

Reviewer #1: Yes

Reviewer #2: Yes

2. Has the statistical analysis been performed appropriately and rigorously? 

Reviewer #1: Yes

Reviewer #2: Yes

3. Have the authors made all data underlying the findings in their manuscript fully available?

Reviewer #1: Yes

Reviewer #2: Yes

4. Is the manuscript presented in an intelligible fashion and written in standard English?

Reviewer #1: No

Reviewer #2: Yes

5. Review Comments to the Author

Reviewer #1: In the present study, the Authors investigated the effect of dental follicle (DF)MSCs in regeneration of glandular tissue in Sjögren’s Syndrome (SS) mouse model.

Although the manuscript is good, some issues need to be addressed.

- The figure 7 is out of focus and must be changed.

- Pages 11 and 12: add the name of university and hospital and change the “XXXXXX”.

- The Discussion section is too long (n#6 pages), dispersive and chaotic. Reduce it focusing on results, limits and novelties of work.

Reviewer #2: In the present study authors investigated the modulatory and regenerative effects of dental follicle mesenchymal stem cells (DFMSCs) in Sjögren’s Syndrome (SS) mouse model. DFMSCs were injected intraperitoneally, or into submandibular, or lacrimal glands. The immunomodulatory effect was investigated by assessing of intracellular cytokine (IFN-g, IL-17, IL-10) secretion in spleen-derived T helper cells, lymphocyte proliferation and B lymphocyte subsets. The regenerative capability was evaluated by Histologic analysis and flow cytometry of submandibular and lacrimal glands.

The improvement of SS manifestations was observed mainly in systemic injection of DFMSCs, while slight recovery was observed in local injections.

The manuscript is well written and the study is interesting. However, some revisions are needed before the paper can be published:

- In materials and method section, the name of dental hospital is missing.

- Scale bars are missing in Figures 7,8,9

- Histological images need to be improved.

6. PLOS authors have the option to publish the peer review history of their article (what does this mean?). If published, this will include your full peer review and any attached files.

Reviewer #1: No

Reviewer #2: No

---

## [Author Response · Author response to Decision Letter 0]

1 Feb 2022

To The Editor-in-Chief,

We thank the reviewers for their valuable contributions to our study. We revised the manuscript as per reviewers’ comments and explained it in detail below. We changed the figure files in “tif.” file format.

Best regards,

Deniz GENÇ

Corresponding author

Journal Requirements:

Response: We edited the manuscript according to the Journal Requirements.

Response: We included the ethics statements in the Methods section and online submission.

Response: The present study is not a retrospective study.

 “This study is funded by THE SCIENTIFIC AND TECHNOLOGICAL RESEARCH COUNCIL OF TURKEY (TUBITAK) with the Project number of 120S178.”

Response: We thank to the Editorial Board.

“This study is funded by THE SCIENTIFIC AND TECHNOLOGICAL RESEARCH COUNCIL OF TURKEY (TUBITAK) with the Project number of 120S178. We thank to Rüştü Kuşcu for his skillful assistance.“

“This study is funded by XXXXXXXX with the Project number of 120S178. We thank to XXXXXX for his skillful assistance.”

“This study is funded by THE SCIENTIFIC AND TECHNOLOGICAL RESEARCH COUNCIL OF TURKEY (TUBITAK) with the Project number of 120S178.”

Response: We thank to the Editorial Board. We removed the Acknowledgements Section, the funding information has been stated as “This study is funded by THE SCIENTIFIC AND TECHNOLOGICAL RESEARCH COUNCIL OF TURKEY (TUBITAK) with the Project number of 120S178. The funders had no role in study design, data collection and analysis, decision to publish, or preparation of the manuscript.” in the funding information section.

Response: The ethics statement is included in the Methods section.

Reviewers' comments:

Reviewer's Responses to Questions

Comments to the Author

1. Is the manuscript technically sound, and do the data support the conclusions?

Reviewer #1: Yes

Reviewer #2: Yes

Response: We thank the reviewers.

2. Has the statistical analysis been performed appropriately and rigorously?

Reviewer #1: Yes

Reviewer #2: Yes

Response: We thank the reviewers.

3. Have the authors made all data underlying the findings in their manuscript fully available?

Reviewer #1: Yes

Reviewer #2: Yes

Response: We thank the reviewers.

4. Is the manuscript presented in an intelligible fashion and written in standard English?

Reviewer #1: No

Reviewer #2: Yes

Response: We thank the reviewers. English editing has been done. 

5. Review Comments to the Author

Reviewer #1: In the present study, the Authors investigated the effect of dental follicle (DF)MSCs in regeneration of glandular tissue in Sjögren’s Syndrome (SS) mouse model.

Although the manuscript is good, some issues need to be addressed.

- The figure 7 is out of focus and must be changed.

- Pages 11 and 12: add the name of university and hospital and change the “XXXXXX”.

- The Discussion section is too long (n#6 pages), dispersive and chaotic. Reduce it focusing on results, limits and novelties of work.

Response 1: Figure 7 was obtained from a fluorescent microscope (Nikon TS-2 FL Trinocular Inverted Microscope 5.9 Megapixel Head DS-Fi3, Japan). The image resolution of the device was 5.9 Megapixels, therefore it cannot be focused by a device further. But, the format of the figures was changed to “tif.” files to increase the resolution of the images.

Response 2: The name of the university and hospital has been mentioned in the main text as “Muğla Sıtkı Koçman University”.

Response 3: The discussion section has been reduced focusing on the results.

Reviewer #2: In the present study authors investigated the modulatory and regenerative effects of dental follicle mesenchymal stem cells (DFMSCs) in Sjögren’s Syndrome (SS) mouse model. DFMSCs were injected intraperitoneally, or into submandibular, or lacrimal glands. The immunomodulatory effect was investigated by assessing of intracellular cytokine (IFN-g, IL-17, IL-10) secretion in spleen-derived T helper cells, lymphocyte proliferation and B lymphocyte subsets. The regenerative capability was evaluated by Histologic analysis and flow cytometry of submandibular and lacrimal glands.

The improvement of SS manifestations was observed mainly in systemic injection of DFMSCs, while slight recovery was observed in local injections.

The manuscript is well written and the study is interesting. However, some revisions are needed before the paper can be published:

- In materials and method section, the name of dental hospital is missing.

- Scale bars are missing in Figures 7,8,9

- Histological images need to be improved.

Response 1: The name of the university and hospital has been mentioned in the main text as “Muğla Sıtkı Koçman University”.

Response 2: The scale bars are added in Figures 7, 8, and 9.

Response 3: Histological images in Figures 7 and 8 were obtained from a fluorescent microscope (Nikon TS-2 FL Trinocular Inverted Microscope 5.9 Megapixel Head DS-Fi3, Japan). The image resolution of the device was 5.9 Megapixels, therefore it cannot be focused by a device further. We focused the images by changing the file format from “jpeg.” To “tif.”. Histological images in Figure 9 were obtained by a light microscope (Olympus BX46, Japan). The format of the figures was changed to “tif.” files to increase the resolution of the images.

---

## [Decision Letter · Decision Letter 1]

2 Mar 2022

PONE-D-21-36398R1DENTAL FOLLICLE MESENCHYMAL STEM CELLS AMELIORATED GLANDULAR DYSFUNCTION IN SJÖGREN'S SYNDROME MURINE MODELPLOS ONE

Dear Dr. Genç,

Thank you for submitting your manuscript to PLOS ONE. After careful consideration, we feel that it has merit but does not fully meet PLOS ONE’s publication criteria as it currently stands. Therefore, we invite you to submit a revised version of the manuscript that addresses the points raised during the review process.

The Authors did not correctly answer to the concerns of a reviewer. They only deleted some sentences.It is important and needed to follow in details all criticisms raised by the referee and highlight in red colour all the amendments.This is the last round and chance they have, therefore they must pay all the attention needed.

We look forward to receiving your revised manuscript.

Kind regards,

Gianpaolo Papaccio, M.D., Ph.D.

Academic Editor

PLOS ONE

Reviewers' comments:

Reviewer's Responses to Questions

**Comments to the Author**

1. If the authors have adequately addressed your comments raised in a previous round of review and you feel that this manuscript is now acceptable for publication, you may indicate that here to bypass the “Comments to the Author” section, enter your conflict of interest statement in the “Confidential to Editor” section, and submit your "Accept" recommendation.

Reviewer #1: (No Response)

Reviewer #2: All comments have been addressed

2. Is the manuscript technically sound, and do the data support the conclusions?

Reviewer #1: Yes

Reviewer #2: (No Response)

3. Has the statistical analysis been performed appropriately and rigorously? 

Reviewer #1: Yes

Reviewer #2: (No Response)

4. Have the authors made all data underlying the findings in their manuscript fully available?

Reviewer #1: Yes

Reviewer #2: (No Response)

5. Is the manuscript presented in an intelligible fashion and written in standard English?

Reviewer #1: No

Reviewer #2: (No Response)

6. Review Comments to the Author

Reviewer #1: The Authors have partially answered all my previous concerns. The Discussion section remains too long. The Authors have only deleted some sentences. They must modify the discussion as previously requested.

Reviewer #2: (No Response)

7. PLOS authors have the option to publish the peer review history of their article (what does this mean?). If published, this will include your full peer review and any attached files.

Reviewer #1: No

Reviewer #2: No

---

## [Author Response · Author response to Decision Letter 1]

9 Mar 2022

To The Editor-in-Chief,

We thank the reviewers for their valuable contribution to our study. We corrected and revised the article as per reviewers’ comments.

Best regards,

Deniz GENÇ

Corresponding Author

Reviewers' comments:

Reviewer's Responses to Questions

Comments to the Author

1. If the authors have adequately addressed your comments raised in a previous round of review and you feel that this manuscript is now acceptable for publication, you may indicate that here to bypass the “Comments to the Author” section, enter your conflict of interest statement in the “Confidential to Editor” section, and submit your "Accept" recommendation.

Reviewer #1: (No Response)

Reviewer #2: All comments have been addressed

Response: We thank the reviewers.

2. Is the manuscript technically sound, and do the data support the conclusions?

Reviewer #1: Yes

Reviewer #2: (No Response)

Response: We thank the reviewers.

3. Has the statistical analysis been performed appropriately and rigorously?

Reviewer #1: Yes

Reviewer #2: (No Response)

Response: We thank the reviewers.

4. Have the authors made all data underlying the findings in their manuscript fully available?

Reviewer #1: Yes

Reviewer #2: (No Response)

Response: We thank the reviewers.

5. Is the manuscript presented in an intelligible fashion and written in standard English?

Reviewer #1: No

Reviewer #2: (No Response)

Response: We thank the reviewers. The language editing was done.

6. Review Comments to the Author

Reviewer #1: The Authors have partially answered all my previous concerns. The Discussion section remains too long. The Authors have only deleted some sentences. They must modify the discussion as previously requested. 

Response: The discussion section is revised by focusing on the main results of the study. Most of the discussions made in comparison with previous studies have been shortened in line with the main results of our study. In the discussion section, the findings of our study were highlighted and revised.

Reviewer #2: (No Response)

7. PLOS authors have the option to publish the peer review history of their article (what does this mean?). If published, this will include your full peer review and any attached files.

Do you want your identity to be public for this peer review? For information about this choice, including consent withdrawal, please see our Privacy Policy.

Reviewer #1: No

Reviewer #2: No

---

## [Decision Letter · Decision Letter 2]

15 Mar 2022

DENTAL FOLLICLE MESENCHYMAL STEM CELLS AMELIORATED GLANDULAR DYSFUNCTION IN SJÖGREN'S SYNDROME MURINE MODEL

PONE-D-21-36398R2

Dear Dr. Genç,

We’re pleased to inform you that your manuscript has been judged scientifically suitable for publication and will be formally accepted for publication once it meets all outstanding technical requirements.

Kind regards,

Gianpaolo Papaccio, M.D., Ph.D.

Academic Editor

PLOS ONE

Additional Editor Comments (optional):

Reviewers' comments:

Reviewer's Responses to Questions

**Comments to the Author**

1. If the authors have adequately addressed your comments raised in a previous round of review and you feel that this manuscript is now acceptable for publication, you may indicate that here to bypass the “Comments to the Author” section, enter your conflict of interest statement in the “Confidential to Editor” section, and submit your "Accept" recommendation.

Reviewer #1: All comments have been addressed

2. Is the manuscript technically sound, and do the data support the conclusions?

Reviewer #1: Yes

3. Has the statistical analysis been performed appropriately and rigorously? 

Reviewer #1: Yes

4. Have the authors made all data underlying the findings in their manuscript fully available?

Reviewer #1: Yes

5. Is the manuscript presented in an intelligible fashion and written in standard English?

Reviewer #1: Yes

6. Review Comments to the Author

Reviewer #1: (No Response)

7. PLOS authors have the option to publish the peer review history of their article (what does this mean?). If published, this will include your full peer review and any attached files.

Reviewer #1: No

---

## [Editor Report · Acceptance letter]

25 Apr 2022

PONE-D-21-36398R2 

Dental follicle mesenchymal stem cells ameliorated glandular dysfunction in Sjögren’s syndrome murine model 

Dear Dr. Genç:

I'm pleased to inform you that your manuscript has been deemed suitable for publication in PLOS ONE. Congratulations! Your manuscript is now with our production department. 

Kind regards, 

on behalf of

Prof. Gianpaolo Papaccio 

Academic Editor

PLOS ONE